# Long COVID Symptom Management Through Self-Care and Nonprescription Treatment Options: A Narrative Review

**DOI:** 10.3390/ijerph22091362

**Published:** 2025-08-29

**Authors:** Preeti Kachroo, Guy Boivin, Benjamin J. Cowling, Will Shannon, Pascal Mallefet, Pranab Kalita, Alexandru M. Georgescu

**Affiliations:** 1Global Category Medical Affairs, Haleon Plc., 1260 Nyon, Switzerland; mallefet@yahoo.fr (P.M.); alexandru.x.georgescu@haleon.com (A.M.G.); 2Department of Pediatric, Faculty of Medicine, Laval University, Quebec, QC G1V 0A6, Canada; guy.boivin@crchudequebec.ulaval.ca; 3School of Public Health, The University of Hong Kong, Pokfulam, Hong Kong SAR, China; bcowling@hku.hk; 4Pinnacle Health Clinic, North Parramatta, NSW 2151, Australia; will@amazinghealth.com.au; 5Global Category Medical Affairs, Haleon Plc., London KT13 0NY, UK; pranbdr@gmail.com

**Keywords:** long COVID, post-acute sequelae of COVID-19, burden of illness, post-COVID condition, nonprescription drug, over-the-counter drug, self-care, conservative therapies, complementary therapies

## Abstract

Many patients experience unique or persistent symptoms several months following the onset of infection with severe acute respiratory syndrome coronavirus 2, the causative agent of COVID-19. While this condition is commonly referred to as long COVID, no universally accepted definition exists; therefore, many patients go underrecognized and underreported. Long COVID can involve almost any major organ system and is characterized by widely heterogeneous persistent or recurrent symptoms including fatigue, headache, cough, dyspnea, chest pain, cognitive dysfunction, anxiety, and depression. In line with the wide array of symptoms, numerous potential underlying pathophysiologic pathways, including viral persistence, prolonged inflammation, autoimmune reactions, endothelial dysfunction, and dysbiosis of the microbiome of the gut, may contribute to the symptomology of long COVID. Therapy is directed at symptomatic control; however, no pharmacologic treatments are specifically approved for the management of symptoms associated with long COVID. Several common symptoms of long COVID may be managed with nonprescription treatments (pharmacologic and nonpharmacologic). The goal of this review is to provide clinicians with a better understanding of long COVID and review the latest recommendations for managing common mild-to-moderate symptoms with nonprescription treatment options.

## 1. Introduction

Severe acute respiratory syndrome coronavirus 2 (SARS-CoV-2), the virus responsible for the COVID-19 pandemic initially identified in China in December 2019, has infected millions of people globally. COVID-19 is a mostly self-contained upper respiratory infection that usually resolves within a few weeks [1,2]; however, a substantial proportion of patients previously infected with SARS-CoV-2 experience ongoing, recurrent, or new symptoms after resolution of the acute phase of infection in a condition commonly referred to as “long COVID” [1,2,3]. Long COVID is also referred to as postacute sequelae of SARS-CoV-2 infection (PASC), post-COVID condition (PCC), postacute COVID syndrome, and chronic COVID syndrome (CCS), with patients sometimes described as “long-haulers” [3,4,5].

Even now, COVID-19 remains prevalent: as of 20 July 2025, the World Health Organization (WHO) reported more than 103 million total confirmed cases of COVID-19 in the United States, 25.1 million in the United Kingdom, and more than 778 million cases globally since the onset of the pandemic [6]. In this post pandemic period, healthcare practitioners (HCPs) continue to encounter patients with long COVID symptoms. Emerging evidence underscores the value of self-care strategies (refers to individual-led actions to manage health and alleviate symptoms using nonprescription treatments, lifestyle adjustments, and complementary approaches) in alleviating these burdens. This narrative review is intended to provide HCPs with the latest information on long COVID and guidance for management of common long COVID symptoms with nonprescription treatment options.

## 2. Methodology

A literature search was conducted using the PubMed database, limited to English-language publications. The search strategy incorporated a combination of keywords and Medical Subject Headings terms, including “long COVID,” “post-COVID,” “nonprescription,” “self-care,” “self-management,” “symptoms,” and “treatment,” to identify relevant clinical studies and review articles.

To supplement the available evidence, additional searches were performed to locate treatment guidelines specific to long COVID. Clinical guidelines, consensus statements, and treatment recommendations were systematically reviewed. Furthermore, a targeted search of ClinicalTrials.gov was conducted using similar terms to identify ongoing registered trials investigating nonprescription interventions for long COVID. Included studies and guidelines focused on nonprescription treatments or self-care and self-management (refers to collaboration between the individual and healthcare provider(s) to achieve good health) strategies for long COVID symptoms. Studies and guidelines without this focus such as prescription therapies or inpatient care were excluded. Findings were synthesized thematically across key symptom domains.

## 3. What Is Known About Long COVID?

### 3.1. Definition

Long COVID is broadly characterized as a multisystemic condition that describes the collection of persistent symptoms following acute COVID infection, which have a variety of etiologies and disease manifestations [7]. The heterogeneous nature of long COVID precludes the development of a standardized definition [8]. However, WHO has defined long COVID as experiencing a range of symptoms 3 months after the initial infection with symptoms that last for at least 2 months and that cannot be explained by an alternative diagnosis [9,10]. The US National Academies of Sciences, Engineering, and Medicine has defined long COVID as an infection-associated chronic condition that occurs after SARS-CoV-2 infection and is present for at least 3 months as a continuous, relapsing and remitting, or progressive disease that affects one or more organ systems [11]. US Centers for Disease Control and Prevention (CDC) and United Kingdom National Institute for Health and Care Excellence (NICE) guidelines recommend that long COVID should be suspected when symptoms that developed during SARS-CoV-2 infection persist or continue to develop from 4 or more weeks after the onset of infection [12,13,14]. In an attempt to standardize the definition of long COVID, and as a framework for further investigation, the US National Institutes of Health’s (NIH) Researching COVID to Enhance Recovery (RECOVER) initiative is developing a patient-reported, symptom-based approach for the diagnosis of long COVID [8].

### 3.2. Prevalence

Estimates of the prevalence of long COVID vary widely depending on the studies and definition [15]. A meta-analysis of 50 studies estimated the pooled global prevalence of long COVID to be 0.43, although the data showed substantial heterogeneity [16]. Two meta-analyses spanning thousands of patients across multiple countries reported between 45% and 80% of patients experienced at least one unresolved symptom persisting for up to 110 days after self-reported or laboratory-confirmed acute SARS-CoV-2 infection [17,18]. Data from the US CDC Household Pulse Survey collected from 20 August through 16 September 2024, showed that approximately 29.8% of adults with prior COVID-19 infection reported experiencing symptoms of long COVID (defined as symptoms lasting 3 or more months after initial viral infection, and that were not present prior to infection) [19]. Results from a US-based survey of more than 16,000 respondents with self-reported test-confirmed COVID-19 infection at least 2 months prior showed that 14.7% reported continued COVID-19 symptoms for more than 2 months after acute illness [15].

### 3.3. Symptoms of Long COVID

The symptoms of long COVID are diverse, numerous, and affect most major organ systems. More than 200 symptoms have been identified [20]. Common symptoms of long COVID include fatigue; shortness of breath or difficulty breathing; memory, concentration, or sleep problems; persistent cough; chest pain; muscle aches; depression/anxiety; and fever (Figure 1) [21,22,23,24]. Patients with long COVID may have overlapping symptoms with other conditions such as myalgic encephalomyelitis/chronic fatigue syndrome, postural orthostatic tachycardia syndrome and other forms of dysautonomia, or mast cell activation syndrome [14].

Symptoms of long COVID can last from weeks to years after the initial infection, and symptom onset can vary over time (Figure 1) [11]. A study of 806 patients with long COVID found that more than 50% failed to improve in illness severity 1.5 years following initial SARS-CoV-2 infection [25]. The results of a meta-analysis with 257,348 patients with COVID-19 determined that the most commonly reported symptoms of fatigue, sleep disorders, and dyspnea persisted from 3 months to more than a year. The most commonly reported symptoms in the 6-to-9-month interval were cough, headache, and loss of taste and smell, while fatigue, dyspnea, myalgia, and sleep disorders persisted beyond the 12-month interval [23]. Similarly, a South Korean series of surveys conducted at 5 to 6 months, 11 to 12 months, and 24 months after symptom onset or COVID-19 diagnosis (N = 132 respondents) reported that fatigue, sleep disturbances (insomnia), and cognitive dysfunction (trouble concentrating, amnesia, depression, or anxiety) were the most commonly reported symptoms persisting at 24 months [24].

Long COVID carries a heavy psychosocial and economic burden [3,4,8]. Results from the Household Pulse Survey showed that approximately 80.1% of patients who had long COVID at the time of the survey reported some degree of activity limitations, and 24.3% reported significant activity limitations [19]. Long COVID can result in patients being out of work, which translates to hundreds of billions in lost annual revenue; in the US, estimates of lost income range from $105 to $430 billion [26,27]. A 2022 report estimated that the total cost of long COVID, including quality of life, lost earnings, and medical care, was $3.7 trillion in the US alone [28].

## 4. Recommendations for Nonprescription Management of Select Long COVID Symptoms

Treatment of long COVID is challenging, owing in part to the wide array of symptoms and the complexity of the underlying pathophysiologic mechanisms. No pharmacologic treatments are specifically indicated for long COVID [20]; few formal guidelines, consensus statements, or treatment recommendations exist to guide symptom management; and minimal data are available to guide treatment decisions [29,30]. NIH and CDC have guideline documents for the treatment of COVID-related symptoms but lack similar guidance for management of long COVID [31,32].

Nonprescription medications for the treatment of cold- and flu-like symptoms have a long track record of safety and effectiveness; however, similar evidence is lacking for treating comparable symptoms due to long COVID [33]. Given the prolonged nature and complexity of long COVID symptoms, ongoing medical supervision and follow-up care are crucial to manage and mitigate the impact on individuals’ health and well-being. Guidance from NICE advised caution with self-management using nonprescription vitamins/supplements because it is unknown if these products are helpful, are harmful, or have no effect on new or ongoing symptoms [12]. CDC noted that, while some nonprescription treatments may be helpful for indications including headache or documented vitamin deficiencies, others may not be, and HCPs should take care to assess efficacy and the potential for harm and drug interactions before advising patients to start a treatment [14]. Nonetheless, recommendations from CDC and WHO as well as other clinical practice and consensus statements suggest that the use of nonprescription medications and vitamin/supplement-based treatments may be beneficial for the symptomatic management of long COVID in some patients [14,34,35,36]. Table 1 provides a summary of suggestions for management of certain long COVID symptoms with nonprescription treatment options, [5,10,12,14,20,33,34,36,37,38,39,40,41,42,43,44,45,46,47,48,49] and a discussion of select long COVID symptoms and potential nonprescription treatment approaches follows.

### 4.1. Fatigue

Chronic fatigue is the most common symptom of long COVID [17,38,50]. Long COVID–related fatigue has been described as debilitating and unrelenting [50] and is characterized by severe exhaustion after minimal physical or mental exertion, sometimes requiring several days of recovery, or persistent tiredness or exhaustion despite having sufficient sleep. For many affected individuals, fatigue may persist for 6 months or longer and can significantly impact the ability to perform activities of daily living [38]. The incidence of fatigue in patients with long COVID ranges from 35% to 45% at 4 weeks, 30% to 77% at 8 weeks, and 16% to 55% at 12 weeks after COVID-19 infection [1]. In one meta-analysis of studies of patients with long COVID, fatigue was reported in 28.4% of patients in a hospitalized cohort, 34.8% of patients in a nonhospitalized cohort, and 25.2% in a mixed (hospitalized and nonhospitalized) cohort [17].

The nonprescription management of fatigue in patients with long COVID should take an individualized approach. WHO conditionally recommends education and training on strategies to conserve energy and pacing, [10] the goal of which is to develop a plan with a titrated approach to physical activity [38]. Such energy conservation strategies aim to avoid overexertion by prioritizing or modifying activities to make them easier [20,38,50]. Relaxation techniques, [10] aromatherapy, [41] establishing good sleep hygiene, [10,38,40] and following a healthy diet [38] may also be helpful in managing fatigue. Nutritional supplements, including vitamins (B12, C, D), fish oil (docosahexaenoic acid/eicosapentaenoic acid [DHA/EPA]), [34,38] and antioxidants, including coenzyme Q10 (CoQ10), may help with fatigue, but more research is needed to better understand their role in long COVID-associated fatigue management [20,38,39].

While supplements are commonly used to address fatigue, their indiscriminate use may pose risks. For instance, vitamin D toxicity, though rare, can result in hypercalcemia and vascular calcification, particularly when used without monitoring serum levels [51]. Similarly, CoQ10, may interfere with anticoagulant therapy (e.g., warfarin), necessitating monitoring. These risks underscore the importance of individualized supplementation based on confirmed deficiencies and clinical oversight [51].

### 4.2. Neurologic

#### 4.2.1. Cognitive Impairment

Up to 28% of patients experience cognitive impairment or trouble concentrating that persists for 1 to 3 months after acute COVID-19 infection, [12] and the risk of cognitive deficits and other neurologic and psychologic disorders (e.g., psychotic disorders, dementia, and epilepsy or seizures) can persist for up to 2 years [52]. Patients describe a feeling of “brain fog” (i.e., a feeling of mental sluggishness, poor attention span, impaired memory), tremors, dizziness, and difficulty concentrating [7,53]. Supportive self-management and symptom monitoring may help manage cognitive symptoms [36]. If cognitive dysfunction is due to an inflammatory process, symptoms may improve on their own with time, or treatment with anti-inflammatory medication may be warranted which may require medical supervision [7]. WHO conditionally recommends self-management including cognitive exercises and implementation of assistive products and environmental modifications as needed, [10] while the United States Department of Veterans Affairs (VA) recommends diaphragmatic breathing [34]. Some research suggests that luteolin, a natural flavonoid, may improve cognitive function, but more research is needed [43,50,54].

#### 4.2.2. Anosmia

Anosmia is another common symptom associated with long COVID [18,55]. Up to 46% of patients reported anosmia at 1 month or longer after COVID-19 infection [12]. Phantosmia (detection of smells that are not present) and dysosmia (altered sense of smell/taste such as excessive chemical, salty, or sour sensations) may also occur [34]. Anosmia typically lasts for a few days, but persistent olfactory dysfunction has been reported at 6 months and 1-year postinfection [56,57]. Although there are few recommended treatments, WHO and VA recommend olfactory retraining as an attempt to regain sense of smell [10,34].

#### 4.2.3. Anxiety, Depression, Psychiatric Illness

Anxiety and depression have been reported in 25% to 30% of people after acute COVID-19 infection [8]. These conditions may be exacerbated by issues such as job loss and the negative impact of long COVID on daily activities and socialization [34]. The emotional and psychological impact of long COVID should not be underestimated, and addressing these consequences is critically important. WHO and VA recommend that patients with long COVID have psychological support offered to them and be instructed on methods of self-care (e.g., guided meditation and stress management) and physical exercise training (e.g., tai chi and yoga) [10,34,58]. Maintaining social interactions and peer support are also important components of mental health and symptom control [10,36]. NICE recommends that patients with suspected long COVID who are experiencing severe psychiatric symptoms or are at high risk of self-harm or suicide should be urgently referred for psychiatric assessment [12].

Nutraceutical supplementation has been studied for the management of psychological sequelae associated with long COVID. The VA suggests that 1000 mg of fish oil (500 mg combined EPA and DHA) may be considered in individuals who are not taking blood thinners or experiencing gastric reflux disease [34].

### 4.3. Pain

#### 4.3.1. Nonspecific Pain

Pain is one of the most commonly reported symptoms in patients with long COVID [57,58] and is described as nonspecific or localized to the joints, back, muscles, chest, feet, skin, abdomen, throat, mouth, and pelvis [8,59]. The true prevalence of long COVID–associated pain can vary widely due to the subjective nature of pain, underreporting, and inconsistent methods of evaluating pain [60]. One meta-analysis showed that chest, gastrointestinal (GI), joint, muscle, general body, and nervous system–related pain symptoms were experienced by approximately 8%, 6%, 18%, 18%, 17%, and 12% of all individuals up to 1 year after recovering from confirmed COVID-19 infection, respectively [60]. In another meta-analysis of patients experiencing unresolved self-reported COVID-19-related symptoms, 27.9% of hospitalized patients reported pain/discomfort and 17.0% of nonhospitalized patients reported muscle pain/myalgia [17]. Other studies have reported persistent myalgia/muscle pain in 4.5% to 51% of patients at 1 to 3 months after hospitalization for COVID-19 infection [5].

As pain related to long COVID may vary in clinical presentation, nonprescription treatment modalities should be used under the guidance of an HCP to ensure the appropriate treatment is administered. Nonprescription analgesics (acetaminophen and nonsteroidal anti-inflammatory drugs [NSAIDs], ibuprofen) can be effective in alleviating arthralgia and myalgia during both acute and postacute phases of COVID-19 infection, but for patients experiencing prolonged pain during the postacute period (i.e., pain persisting for several months), self-management is not encouraged. Instead, these patients should be referred to a specialist or multidisciplinary team for tailored treatment to avoid possible adverse events associated with long-term analgesic use [61]. For joint pain, WHO recommends pain education, self-management skills training, short-term use of anti-inflammatory agents, and physical exercise training for patients without post exertional symptom exacerbation (PESE) [10]. In case studies, analgesics (NSAIDs, paracetamol, diclofenac, and oral calcium) have resolved joint pain within 1 to 2 weeks [62]. Providers may consider these agents for short-term symptomatic relief; however, their off-label or indiscriminate use must be avoided in the absence of appropriate clinical oversight and judgement. Limited research concluded that Pilates, resistance exercises, and neuromodulation may benefit some patients with arthralgia, but more research is needed to understand which patients may benefit from these interventions [63].

#### 4.3.2. Headache

Headache is one of the most commonly reported neurologic symptoms in patients with long COVID [64]. Approximately 40% of patients reported experiencing headache following COVID-19 infection [8]. Headache can present as a new headache phenotype or exacerbation in those with a history of headache, or as a new-onset headache syndrome in those without a history of headache [47]. Long COVID–associated headache has been characterized as typically bilateral, frontal, or periocular in location and oppressive in severity; however, presentation varies widely. The most common headache phenotypes are tension-type and migraine-like headaches [64].

First-line treatment recommendations for long COVID–related headache include abortive analgesics (acetaminophen and NSAIDs), although these treatments may not achieve a full response [47]. Treatment of headache according to its closest phenotype (e.g., migraine) can guide secondary treatment, but most require a prescription under medical supervision and have been found in case studies to be largely ineffective [47]. The VA recommends lifestyle management (adequate sleep, hydration, exercise, and nutrition; keeping a headache diary; maintenance of glucose levels), and daily supplementation with riboflavin 400 mg and magnesium oxide 420 mg [34].

#### 4.3.3. Flu-like Symptoms

Flu-like symptoms, such as sore throat, sneezing, or a blocked, stuffy, or runny nose, may occur with long COVID. These symptoms typically have a short duration and can be managed with rest, hydration, and nonprescription treatments similar to other viral illnesses [33]. Providers could consider paracetamol, NSAIDs, cough suppressants, or other nonprescription cold and flu treatments for these patients [33].

### 4.4. Respiratory

#### 4.4.1. Cough

The percentage of patients with long COVID and cough varies widely. Results from one study showed that 15% to 81% of hospitalized and approximately 12% of nonhospitalized patients reported experiencing persistent cough 1 to 12 months after COVID-19 infection [65]. In a meta-analysis, rates ranged from 34% to 60%, and in other studies, rates hovered around 30% [5,8,65,66].

Nonpharmacologic treatment of cough includes breathing control exercises [36,46] and airway clearance techniques [44]. The VA recommends diaphragmatic breathing and sputum management (hydration, expectorants, airway clearance devices) [34]. Nonprescription medications commonly used to treat cold- and flu-like symptoms, including analgesics and cough suppressants, [33] and proton pump inhibitors (PPIs) for patients with reflux, [36] may also be of benefit in appropriately selected patients with long COVID [36]. Long COVID sufferers with worsening cough or cough persisting longer than 12 weeks despite treatment should seek specialist advice for pulmonary testing [34].

#### 4.4.2. Dyspnea

Persistent breathing issues can be severe and disabling and negatively impact quality of life [44,67]. In one study, up to 43.4% of patients had persistent dyspnea 60 days after COVID-19 infection [68]. In a small study (N = 76), dyspnea was present in 49% of patients at 3 months and 46% of patients at 12 months after initial COVID-19 infection [69]. In this study, 80% of individuals who were dyspneic at month 12 reported that dyspnea had persisted from month 3, and 20% reported developing new-onset clinically meaningful dyspnea at 3 to 12 months postinfection [69]. Underlying and coexisting issues causing dyspnea may include cardiovascular complications, anxiety/depression, anemia, dysautonomia, and posttraumatic stress disorder. Clinical guidance from the PASC Collaborative (from the American Academy of Physical Medicine and Rehabilitation) recommends considering these conditions when suggesting a diagnostic test or treatment [44].

Nonpharmacologic interventions for self-managed dyspnea alleviation include breathing exercises, pulmonary rehabilitation, and adjusting body position [44,50]. Pulmonary rehabilitation may include medical management, low-intensity exercise, behavior and occupational modification, and psychosocial and vocational support [48]. WHO conditionally recommends education and skills training for self-management, including nasal breathing, pacing, and exercising (for patients without PESE) [10]. The VA recommends a heart-healthy diet, stress management, and diaphragmatic breathing [34]. While symptoms may improve with supportive care, unresolved or severe cases may require referral to specialist cardiopulmonary services. Multidisciplinary long COVID clinics are usually recommended when breathlessness coexists with other symptoms or when diagnostic uncertainty persists [70].

### 4.5. Gastrointestinal

Results from a large systematic literature review that included more than 735,000 patients with long COVID (average follow-up time of approximately 4 months) found that gastrointestinal (GI) symptoms, including nausea/vomiting, constipation, diarrhea, and stomach pain, occurred in 11.7% of patients, including 6.4% of hospitalized patients and 7.7% of nonhospitalized patients [17]. In a survey study nested within a prospective cohort of 1783 individuals with at least 6 months of follow-up after acute COVID-19 infection, 29% of 749 respondents self-reported having GI symptoms 6 months after the diagnosis of COVID-19. These symptoms included heartburn (16.0%), constipation (11.0%), diarrhea (9.6%), abdominal pain (9.4%), and nausea/vomiting (7.1%) [71]. In a summary of digestive complaints in patients with long COVID, weight loss of greater than 5%, diarrhea or vomiting, diarrhea, and anorexia were the most commonly reported GI symptoms occurring in 17%, 11%, 9%, and 8%, respectively, at 8 weeks following acute COVID-19 infection [1].

Many GI symptoms may be responsive to treatment with nonprescription and nonpharmacologic options. For example, prebiotics (such as lactoferrin 400–600 mg/day) and probiotics may be helpful in the management of gut dysbiosis; dietary supplements, including the flavonoids luteolin and quercetin, may have immunomodulatory and anti-inflammatory properties; and PPIs can be useful for acid reflux [4,20,36,72].

While self-care strategies may offer symptomatic relief, their unsupervised or off-label use should be avoided. For instance, probiotics may pose a rare risk of bacteremia or fungemia in immunocompromised individuals, and prolonged or high-dose PPI use may lead to Clostridioides difficile infection, and rebound acid hypersecretion, particularly in those with complex or persistent symptoms [73].

## 5. Ongoing Research

Long COVID is the subject of extensive international research efforts. The NIH RECOVER initiative is a program aiming to characterize the clinical spectrum, etiology, natural history, and prevalence of long COVID [30]. The National Research Action Plan on Long COVID is the United States Department of Health and Human Services’ plan for promoting research in long COVID [3]. The University of Birmingham is conducting the Therapies for Long COVID (TLC) study in nonhospitalized individuals, from symptoms, patient-reported outcomes, and immunology to targeted therapies [74]. In addition, several commercial research programs are currently exploring nonprescription treatments for symptoms associated with long COVID (Table 2). Current research is also investigating whether vaccination against SARS-CoV-2 is an effective means to prevent long COVID [75].

## 6. Limitations

This literature review on long COVID, while comprehensive, acknowledges certain limitations. While every effort has been made to include all available relevant information, the included studies are limited to those available in PubMed and written in English; it is possible some reports have been missed from other databases such as EMBASE or Scopus and non-English sources. This limitation could affect the generalizability of the findings, particularly in regions where research is published in local languages or indexed outside PubMed. Nonetheless, PubMed provided robust coverage of peer-reviewed biomedical literature, and the inclusion of clinical guidelines and consensus statements helped address gaps in empirical data. Moreover, for some interventions, only a small number of publications are available and limit our understanding of their safety and effectiveness. Furthermore, long COVID is characterized by a broad spectrum of symptoms that may not be fully represented in this review.

Long COVID remains a relatively new and complex medical entity, and our understanding of the disease and its management are constantly evolving [76,77]. As new insight is undoubtedly gained, some recommendations included in this review may be strengthened or weakened.

## 7. Conclusions

Long COVID is a complex, multisystemic condition that places a heavy medical burden on patients, but also on society as a whole [7]. Beyond the direct impact on the healthcare system, indirect effects can include reduced workforce productivity, increased need for caregiver support, and the mental health implications for patients dealing with persistent symptoms. Subsequent economic implications related to lost earnings and requirement for ongoing care are likely to be significant. As such, addressing long COVID requires a comprehensive response, encompassing healthcare, social support systems, and economic policy.

No treatments are currently approved specifically for long COVID, but this review outlines recommended approaches for managing individual symptoms. Several effective nonprescription options are generally recommended for symptomatic long COVID management, including pacing, supplements, and lifestyle changes for fatigue; NSAIDs and acetaminophen for pain, fever, and sore throat; cognitive and breathing exercises for neurocognitive and respiratory symptoms; olfactory training for anosmia; psychological support and meditation for mood disorders; dietary supplementation including probiotics for gastrointestinal symptoms; and NSAIDs, supplements, and antioxidants for chronic inflammation.

Healthcare professionals are encouraged to apply a symptom-oriented, multidisciplinary approach, tailoring interventions to individual patient needs while remaining attentive to emerging evidence. Facilitating access to validated resources and fostering patient engagement in self-care strategies may further improve outcomes. Future research will enhance our understanding of the pathogenesis and natural history of long COVID and will further establish the efficacy of nonprescription treatments, ultimately guiding the development of therapies that target root causes.

## Figures and Tables

**Figure 1 ijerph-22-01362-f001:**
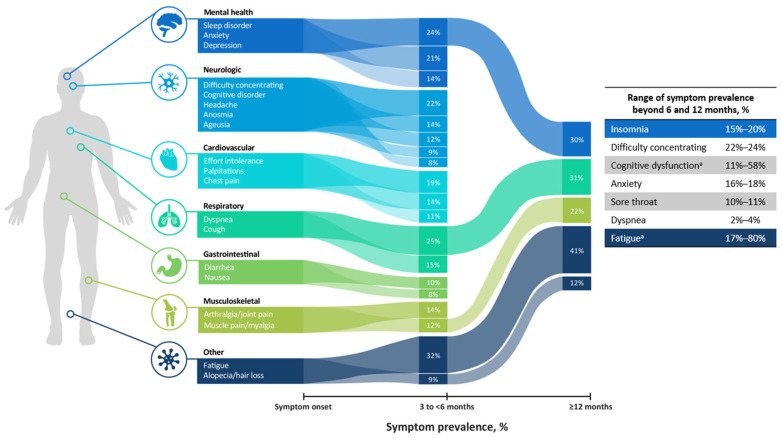
Prevalence, timing, and duration of commonly reported symptoms of long COVID at 3 to <6 months and ≥12 months (data in figure on left from Alkodaymi 2022, data for ranges of symptom prevalence beyond 6 and 12 months from Kim 2023 and Davis 2021) [22,23,24]. Where, ^a^: Percentages for fatigue and cognitive dysfunction combine those from Kim 2023 and Davis 2021 and include symptom prevalence at 6–7 months (Davis 2021) and 12 and 24 months (Kim 2023) postacute COVID infection. All others are ranges at 12 and 24 months postacute COVID-19 infection (Kim 2023) [22,24].

**Table 1 ijerph-22-01362-t001:** Nonprescription Treatment Options for Patients with Long COVID Symptoms.

Symptom	Recommendation	Type of Support/Sources
**General signs and symptoms**		
Fatigue	Convalescence	ME Association [37]
	Energy conservation and pacing	WHO [10] ME Association
		AAPM&R [38]
	Dietary supplements, vitamins (B12, C, D), fish oil (DHA/EPA) 1000 mg	VA [34] ME Association [37]
		AAPM&R [38]
		Davis et al. [20]
	CoQ10	AAPM&R [38]
		Wood et al. [39] Davis et al. [20]
	Lifestyle modification, healthy diet, hydration, sleep	AAPM&R [38] AAPM&R [40]
	Aromatherapy (blend of essential oils from thyme, orange peel, clove bud, and frankincense)	Hawkins et al. [41]
Fever	Paracetamol, NSAIDs	Greenhalgh et al. [36]
**Neurologic**		
Memory, concentration, attention, language, sleep issues	Self-management, cognitive exercises, attention and memory strategies, supportive management, diaphragmatic breathing	WHO [10] VA [34]
		AAPM&R [42]
	Cognitive pacing	Davis et al. [20]
	Word-finding and comprehension strategies	AAPM&R [42]
	Luteolin	Theoharides et al. [43]
	Symptom monitoring	Greenhalgh et al. [36]
Anosmia	Olfactory training	VA [34] WHO [10]
Depression/anxiety	Vitamins, electrolyte supplements	CDC [14]
	Fish oil	VA [34]
	Psychological support	WHO [10] VA [34] NICE [12]
		Greenhalgh et al. [36]
	Mindfulness-based approaches (stress reduction, meditation)	WHO [10]
	Peer support groups	WHO [10] VA [34]
		Greenhalgh et al. [36]
	Exercise training	WHO [10] VA [34]
	Self-care (yoga, tai chi)	WHO [10] VA [34]
		Greenhalgh et al. [36]
**Respiratory**		
Shortness of breath	Self-management, breathing exercises, exercise training, and pacing	WHO [10] CDC [14] VA [34]
		AAPM&R [44]
		Mikkelsen and Abramoff [45]; Akbarialiabad et al. [5]
	Pulmonary rehabilitation exercises	VA [34]
		AAPM&R [44]
	Posture improvement	AAPM&R [44]
	Heart-healthy diet	VA [34]
Persistent cough	Self-directed breathing exercises and airway clearance techniques	VA [34]
		AAPM&R [44]
		Greenhalgh et al. [36]
	Cough suppressants (e.g., benzonatate, guaifenesin, dextromethorphan, gabapentin, amitriptyline, lidocaine, beta-agonists, leukotriene antagonists)	Mikkelsen and Abramoff [45]; Rai et al. [46]
	Excess sputum management (expectorants, hydration, airway clearance techniques)	VA [34]
		AAPM&R [44]
**Pain**		
Headache	Nonprescription analgesics such as acetaminophen (paracetamol)	VA [34]
		AAPM&R [40]
		Chhabra et al. [47]
	Electrolyte supplements, melatonin	AAPM&R [40]
	Hydration	VA [34]
	Maintenance of glucose levels	VA [34]
	Lifestyle management (regular sleep, meals, hydration, and exercise with stress management)	VA [34]
		AAPM&R [40]
	Supportive management and symptom monitoring	Greenhalgh et al. [36]
Muscle/joint pain	Pain education and self-management	WHO [10]
	Anti-inflammatory agents (NSAIDs) and analgesics (acetaminophen or ibuprofen, trolamine salicylate, diclofenac)	WHO [10]
		AAPM&R [40]
Chest pain	NSAIDS (ibuprofen)	VA [34]
		Mikkelsen and Abramoff [45]
Sore throat	Paracetamol, aspirin, ibuprofen	Eccles et al. [33]
**Other**		
Gastrointestinal symptoms	Probiotics	Davis et al. [20]
	Dietary supplementation	Stanford Hall (United Kingdom) Consensus Statement for Post-COVID Rehabilitation [48]
	Herbal medicine with/without acupuncture	Gawey et al. [49]
Chronic inflammation	NSAIDs, CoQ10, and antioxidants	Akbarialiabad et al. [5]

AAPM&R: American Academy of Physical Medicine and Rehabilitation; CDC: Centers for Disease Control and Prevention; CoQ10: coenzyme Q10; DHA: docosahexaenoic acid; EPA: eicosapentaenoic acid; ME: Myalgic Encephalomyelitis; NICE: National Institute for Health and Care Excellence; NSAID: nonsteroidal anti-inflammatory drug; VA: United States Department of Veterans Affairs; WHO: World Health Organization. Shading key: green shading = guideline or government-issued guidance; blue shading = consensus statement; orange shading = clinical trial or review; purple shading = complementary medicine.

**Table 2 ijerph-22-01362-t002:** Recent and Ongoing Clinical Trials Assessing Nonprescription Treatment Options for Long COVID.

NCT	Intervention	Study Size	End Date/Anticipated End Date	Sponsor	Primary Outcome
NCT04810065	Twice-weekly singing and breathing retraining intervention conducted over 10 weeks	30 patients with lung issues due to COVID-19 including breathing difficulties, shortness of breath, and/or reduced exercise tolerance	1 September 2021	University of Limerick	12-week COVID-19 Yorkshire Rehabilitation Scale (C19-YRS)
NCT04841759	Physical exercise–based rehabilitation program	46 healthcare workers with and without post–COVID-19 fatigue	22 December 2021	Medical University of Vienna	Change of maximum oxygen uptake (VO_2_max) from baseline to 4 weeks and 8 weeks
NCT04809974	Niagen (nicotinamide riboside)	70 participants with long COVID	23 February 2024	Massachusetts General Hospital	Executive functioning and memory composite scores at baseline vs. 12 and 22 weeks
NCT04813718	Omni-Biotic Pro Vi 5 (probiotic)	20 participants with long COVID	7 December 2024	Medical University of Graz	Microbiome composition, multiple gut signaling biomarkers, and lung function tests
NCT05121766	Omega-3 (EPA + DHA)	100 healthcare workers with long COVID	21 April 2023	Hackensack Meridian Health	Feasibility study: levels of compliance, interest, and completion
NCT04960215	Coenzyme Q10	121 participants with long COVID	10 February 2022	Aarhus University Hospital	Self-reported symptoms and symptom scores

DHA: docosahexaenoic acid; EPA: eicosapentaenoic acid.

## Data Availability

Not applicable.

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
