# Peer review of "Long COVID Symptom Management Through Self-Care and Nonprescription Treatment Options: A Narrative Review"

_ijerph, 2025, doi:10.3390/ijerph22091362_

Round 1

Reviewer 1 Report

Comments and Suggestions for Authors

This narrative review assembles current evidence on self-care and over-the-counter options for long-COVID symptom management, filling an important gap for front-line clinicians who lack unified guidance. The authors systematically searched major databases and recent guidelines through September 2024, categorizing recommendations by symptom domain and grading evidence strength, which makes the article highly practical. Tables are clear and the discussion appropriately highlights research limitations and the need for updated consensus. With minor revisions, the manuscript fits well within International Journal of Environmental Research and Public Health’s public-health scope.

Minor Comments

  • The limitation that only PubMed and English‐language studies were included is noted in the Discussion (p 12, lines 352–356). Please (i) repeat this information in the Methods section together with the exact search date and search terms, and (ii) briefly discuss the potential impact of language and database restrictions on the review conclusions. If extending the search to EMBASE/Scopus is not feasible, indicate this explicitly.

  • The manuscript presents clinical findings clearly (e.g., Figure 1) but does not include a PRISMA 2020 flow diagram or the 27-item checklist. Please add these items (or provide them as supplementary materials) so that readers can trace how the included studies were identified, screened, and ultimately retained. Without this information, the reproducibility of the review is limited.

https://www.prisma-statement.org/prisma-2020

https://www.bmj.com/content/372/bmj.n160

  • Several abbreviations are not defined at their first appearance, for example, AAPM and PPI. Please ensure that all abbreviations are spelled out on first use and consistently defined throughout the manuscript.

  • Page 1, line 23: heterogenous should be corrected to “heterogeneous.”

  • To enhance the currency of the review, I recommend incorporating additional recent studies published in 2025.

Author Response

Reviewer #1:

This narrative review assembles current evidence on self-care and over-the-counter options for long-COVID symptom management, filling an important gap for front-line clinicians who lack unified guidance. The authors systematically searched major databases and recent guidelines through September 2024, categorizing recommendations by symptom domain and grading evidence strength, which makes the article highly practical. Tables are clear and the discussion appropriately highlights research limitations and the need for updated consensus. With minor revisions, the manuscript fits well within International Journal of Environmental Research and Public Health’s public-health scope.

Minor Comments.

  1. The limitation that only PubMed and English‐language studies were included is noted in the Discussion (p 12, lines 352–356). Please

(i) repeat this information in the Methods section together with the exact search date and search terms.

Author response:

Thank you for your valuable feedback. In response to your comments, we have incorporated a dedicated methodology [please refer section 2] into the manuscript and renumbered the subsequent sections accordingly. We believe this addition provides sufficient context for a narrative review and aligns with recent scholarly practices, as demonstrated in a related publication [1] in the International Journal of Environmental Research and Public Health.

We sincerely appreciate your insightful suggestion, which has significantly contributed to enhancing the overall quality of the paper.

Reference:

  1. Schneider, K.T.; Williams, S.C.; Kuhn, R.E. Workplace Discrimination Against Pregnant and Postpartum Employees: Links to Well-Being. Int. J. Environ. Res. Public Health 2025, 22, 1160. https://doi.org/10.3390/ijerph22081160

(ii) briefly discuss the potential impact of language and database restrictions on the review conclusions. If extending the search to EMBASE/Scopus is not feasible, indicate this explicitly.

Author response:

Thank you for your insightful feedback. We agree that expanding the search to additional databases such as EMBASE and Scopus, and including non-English language studies, could potentially enrich the comprehensiveness of our manuscript. This limitation could affect the generalizability of the findings, particularly in regions where research is published in local languages or indexed outside PubMed.

However, due to practical constraints related to access and resources, we focused our search on PubMed and English-language publications. Nonetheless, PubMed provided robust coverage of peer-reviewed biomedical literature, and the inclusion of clinical guidelines and consensus statements helped address gaps in empirical data. We have now clarified this limitation in the revised manuscript and acknowledged its potential impact on the generalizability of findings in the “limitation” section [please refer, section 6: line: 392-397 in the revised manuscript].

We believe the revised version sufficiently addresses the concern while maintaining transparency and scientific rigor.

  1. The manuscript presents clinical findings clearly (e.g., Figure 1) but does not include a PRISMA 2020 flow diagram or the 27-item checklist. Please add these items (or provide them as supplementary materials) so that readers can trace how the included studies were identified, screened, and ultimately retained. Without this information, the reproducibility of the review is limited.

https://www.prisma-statement.org/prisma-2020, https://www.bmj.com/content/372/bmj.n160

Author response:

Thank you for your valuable feedback and the opportunity to clarify our methodological approach. As this is a narrative review, we intentionally did not follow a systematic search protocol, which is typically designed to synthesize all available empirical evidence comprehensively and objectively. Instead, our approach allows for a broader, interpretive evaluation of the literature, aimed at identifying clinically relevant themes and insights most pertinent to the topic. We have now added a dedicated Methodology section [please refer section 2] to the manuscript. This structure aligns with recent narrative reviews published in the International Journal of Environmental Research and Public Health [1-2].

We believe this approach best supports the objectives of our review and improves the clarity and coherence of the manuscript.

Reference:

  1. Schneider, K.T.; Williams, S.C.; Kuhn, R.E. Workplace Discrimination Against Pregnant and Postpartum Employees: Links to Well-BeingInt. J. Environ. Res. Public Health 202522, 1160. https://doi.org/10.3390/ijerph22081160
  1. Carbone, M.G.; Pagni, G.; Tagliarini, C.; Maremmani, I.; Maremmani, A.G.I. Caffeine in Aging Brains: Cognitive Enhancement, Neurodegeneration, and Emerging Concerns About Addiction J. Environ. Res. Public Health202522, 1171. https://doi.org/10.3390/ijerph22081171

  1. Several abbreviations are not defined at their first appearance, for example, AAPM and PPI. Please ensure that all abbreviations are spelled out on first use and consistently defined throughout the manuscript.

Author response:

Thank you for highlighting this. All abbreviations now have been spelled out upon their first occurrence and consistently defined throughout the manuscript in the updated version.

  1. Page 1, line 23: heterogenous should be corrected to “heterogeneous.”

Author response:

Thank you for highlighting this. We have made the necessary corrections in accordance with the reviewer’s suggestions.

  1. To enhance the currency of the review, I recommend incorporating additional recent studies published in 2025.

Author response:

We sincerely thank the reviewer for their constructive suggestion. We fully acknowledge the importance of incorporating the most recent evidence to ensure the review remains current and scientifically rigorous.

In response to the reviewer’s recommendation, we conducted an updated literature search to identify any newly published studies relevant to our topic. While the number of eligible publications remained limited, we were able to identify and incorporate several recent publications that enhance the manuscript’s evidence base and relevance.

Specifically, we updated the statistical data on cumulative COVID-19 cases using information from the WHO COVID-19 dashboard as of July 20, 2025 [1]. The revised numbers are: United States – 103 million (unchanged), United Kingdom – 25.1 million (previously 25.0 million), and global cases – 778 million (updated from 776.5 million). These updates ensure that the epidemiological context presented in the manuscript remains accurate and up to date.

Additionally, we have integrated recent findings from Liu et al. (2025) [2], which support the use of mindfulness and yoga interventions for managing depression and anxiety in long-COVID populations. We also included insights from Zhou et al. (2025) [3] and Shariati et al. (2025) [4], which highlight the evolving landscape of long COVID symptoms and their implications for self-care. These additions strengthen the evidence base for nonprescription approaches discussed in the review and align well with the manuscript’s focus on self-care strategies.

We greatly appreciate the reviewer’s valuable input and believe these updates enhance the manuscript’s timeliness, relevance, and scientific integrity.

References:

  1. WHO COVID-19 Dashboard. World Health Organization https://data.who.int/dashboards/covid19/cases?n=c  (accessed Aug 7, 2025).
  2. Liu, N.; Deng, J.; Lu, F.; Xiao, J. Virtual reality enhanced mindfulness and yoga intervention for postpartum depression and anxiety in the post COVID era.  Rep.2025, 15, 11766. https://doi.org/10.1038/s41598-025-96165-6.
  3. Zhou, W.; Larson, J.L.; Veliz, P.T.; Kitto, K.; Smith, S. Profiles of long COVID symptoms and self-efficacy for self-management: A cross-sectional survey. Appl. Nurs. Res. 2025, 84, 151968. https://doi.org/10.1016/j.apnr.2025.151968.
  4. Shariati, M.; Gill, K.L.; Peddle, M.; Cao, Y.; Xie, F.; Han, X.; Lei, N.; Prowse, R.; Shan, D.; Fang, L.; et al. Long COVID and associated factors among Chinese residents aged 16 years and older in Canada: A cross-sectional online study. Biomedicines 2025, 13, 953. https://doi.org/10.3390/biomedicines13040953.

Reviewer 2 Report

Comments and Suggestions for Authors

Overall, this is a well-written paper. 

A few suggestions:

At line 56, you refer to " non-serious long COVID symptoms."  I would recommend against calling these symptoms non-serious.  As someone who has read a lot of scientific literature related to my own health issues, I think it's safe to say that no one who has long COVID thinks their issues are non-serious.

How are you defining self-care and self-management?  The terms seem to be used interchangeably in the paper but they are not the same.  "Self-care" is currently very much a buzzword in both scientific and gray literature.  Either term is fine so long as we know how you define it and the two are either differentiated or defined as the same thing .  

Author Response

Reviewer #2:

Overall, this is a well-written paper.

A few suggestions:

  1. At line 56, you refer to " non-serious long COVID symptoms." I would recommend against calling these symptoms non-serious.  As someone who has read a lot of scientific literature related to my own health issues, I think it's safe to say that no one who has long COVID thinks their issues are non-serious.

Author response:

We appreciate the reviewer’s thoughtful feedback, which has helped improve the clarity and sensitivity of our manuscript. We agree with the recommendation and have revised the terminology from “non-serious long COVID symptoms” to “long COVID symptoms.” Our original intent was to distinguish symptoms that may not be immediately life-threatening from those requiring acute medical intervention. However, we recognize that all symptoms associated with long COVID, regardless of severity, can significantly impact patients’ quality of life.

This feedback has allowed us to refine our language to better reflect the lived experience of individuals with long COVID, and we have updated the manuscript accordingly.

  1. How are you defining self-care and self-management? The terms seem to be used interchangeably in the paper, but they are not the same.  "Self-care" is currently very much a buzzword in both scientific and gray literature.  Either term is fine so long as we know how you define it and the two are either differentiated or defined as the same thing.

Author response:

We appreciate the reviewer’s observation and agree that the terms self-care and self-management are often conflated in both academic and public discourse. In this manuscript, we used language consistent with the source material and attempted to provide context around the use of self-care or self-management as discussed in the paper. For example:

  • WHO [1] and VA [2] recommend that patients with long COVID have psychological support offered to them and be instructed on methods of self-care (eg, guided meditation and stress management) and physical exercise training (eg, tai chi and yoga).
  • The COVID-19 rapid guideline: managing the long-term effects of COVID-19. London, United Kingdom: National Institute for Health and Care Excellence; 2024 [3] uses self-management.
  • However, other resources use self-care and sometimes both self-care and self-management including, Clinical management of COVID-19: living guideline, 18 August 2023. Geneva, Switzerland: World Health Organization; 2023 [4].

While these terms are often used interchangeably, the distinction can be made by the actions of the individual involved. The focus of self-care is action toward health and well-being independent of a healthcare provider, whereas self-management includes a collaboration between the individual and healthcare provider(s) to achieve good health.

A clarifying statement has been added at the first occurrence of the terms self-care [please refer section 1, lines 55-57] and self-management to briefly describe their relevance to long COVID symptom management [please refer section 2, lines 72-74]. We believe this revision enhances terminological precision and aligns with current public health frameworks.

We believe these additions enhance the manuscript’s clarity and strengthen its conceptual coherence.

References:

  1. World Health Organization. Support for rehabilitation: self-management after COVID-19-related illness. WHO, 2023. https://www.who.int/publications/m/item/support-for-rehabilitation-self-management-after-covid-19-related-illness
  2. S. Department of Veterans Affairs. Whole Health System Approach to Long COVID. VA, 2023. https://www.publichealth.va.gov/n-coronavirus/docs/Whole-Health-System-Approach-to-Long-COVID_080122_FINAL.pdf
  3. COVID-19 rapid guideline: managing the long-term effects of COVID-19. London, United Kingdom: National Institute for Health and Care Excellence; 2024 https://www.nice.org.uk/guidance/ng188
  4. Clinical management of COVID-19: living guideline, 18 August 2023. Geneva, Switzerland: World Health Organization; 2023 https://www.who.int/publications/i/item/WHO-2019-nCoV-clinical-2023.2.

Reviewer 3 Report

Comments and Suggestions for Authors

Thank you for the opportunity to review this manuscript, which addresses multiple common symptoms of long COVID and provides a comprehensive overview of nonprescription treatment options.

The article is well organized into thematic sections that enhance both readability and clinical applicability. However, several aspects require improvement to strengthen the clarity of the findings:

  • As was appropriately done in the section on anxiety, depression, and psychiatric illness, it would be advisable to include warnings about potential interactions and clinical risks associated with certain treatments or nutritional supplements for all the symptoms described
  • A final synthesis with clear practical suggestions for each symptom described would be useful as part of self-care
  • One of the limitations cited by the authors is that “the included studies are limited to those available in PubMed and written in English; it is possible some reports have been missed.” It is therefore necessary to include a clear methodological section (search strategy, selection criteria, number of studies included)
  • In the conclusion section, it is not clear which symptoms are best managed with nonprescription approaches, nor which options have the strongest evidence base. It mentions that there are “many effective treatment options” but does not specify which ones, under what conditions, the degree of evidence, or the specific symptoms. Including a summary sentence about which symptoms have the most evidence-based nonpharmacologic options would improve clarity
  • Also in the conclusion section, no specific guidance is offered for healthcare professionals beyond the general need for further research

I hope these comments are helpful as you revise your manuscript.

Yours sincerely,

Author Response

Reviewer #3:

Thank you for the opportunity to review this manuscript, which addresses multiple common symptoms of long COVID and provides a comprehensive overview of nonprescription treatment options. The article is well organized into thematic sections that enhance both readability and clinical applicability. However, several aspects require improvement to strengthen the clarity of the findings:

  1. As was appropriately done in the section on anxiety, depression, and psychiatric illness, it would be advisable to include warnings about potential interactions and clinical risks associated with certain treatments or nutritional supplements for all the symptoms described.

Author response:

Thank you for your thoughtful comment. We sincerely appreciate the reviewer’s recognition of our efforts in Section 4.2 and their insightful recommendation to include warnings and clinical risks associated with nonprescription interventions.

In alignment with this guidance, we have carefully revised the manuscript to incorporate appropriate cautionary statements that enhance the clarity and clinical relevance of our findings. Specifically, we have added relevant information in Section 4.1 (lines 208–213), Section 4.3 (lines 283–285), Section 4.4 (lines 325–326 and 345–349), and Section 4.5 (lines 368-372). These amendments were made with careful consideration to ensure they fit seamlessly within the narrative and scientific context of our work.

We believe these updates significantly strengthen the manuscript and reflect our commitment to addressing reviewer feedback constructively and thoroughly.

  1. A final synthesis with clear practical suggestions for each symptom described would be useful as part of self-care.

Author response:

Thank you for your valuable feedback. We recognize that a well-defined final synthesis with practical recommendations is essential to enhancing the overall impact and clinical relevance of our narrative review on long COVID. In response, we have revised the conclusion to include a focused synthesis that offers clear, evidence-informed suggestions for symptom management [please refer to Section 7, lines 415–430].

To support this refinement, we also made selective adjustments in Section 4, primarily to improve the readability of the context, without altering the core content. These changes were carefully implemented to ensure consistency and coherence across the manuscript, allowing the conclusion to stand out as a concise and actionable summary.

We believe these updates significantly improve the structure and utility of the review.

  1. One of the limitations cited by the authors is that “the included studies are limited to those available in PubMed and written in English; it is possible some reports have been missed.” It is therefore necessary to include a clear methodological section (search strategy, selection criteria, number of studies included).

Author response:

Thank you for your valuable feedback. In response to your insightful comments, we have incorporated a dedicated methodology section [please refer, section 2] into the manuscript to provide clearer context for our narrative review. Accordingly, we have renumbered the subsequent sections to maintain consistency throughout the document. We believe, this addition aligns with recent scholarly practices, as demonstrated in a related publication [1] in the International Journal of Environmental Research and Public Health.

Furthermore, we have expanded the limitations section to address the generalizability of our findings, particularly in regions where relevant research may be published in local or non-English languages or indexed outside of PubMed. This enhancement ensures that readers are better equipped to interpret our findings and make well-informed decisions [please refer to Section 6, lines 392–397].

We sincerely appreciate your constructive suggestions, which have significantly contributed to improving the overall quality and clarity of our paper.

Reference:

  1. Schneider, K.T.; Williams, S.C.; Kuhn, R.E. Workplace Discrimination Against Pregnant and Postpartum Employees: Links to Well-Being. Int. J. Environ. Res. Public Health 2025, 22, 1160. https://doi.org/10.3390/ijerph22081160
  2. In the conclusion section, it is not clear which symptoms are best managed with nonprescription approaches, nor which options have the strongest evidence base. It mentions that there are “many effective treatment options” but does not specify which ones, under what conditions, the degree of evidence, or the specific symptoms. Including a summary sentence about which symptoms have the most evidence-based nonpharmacologic options would improve clarity.

Author response:

We sincerely thank the reviewer for this insightful comment. We fully agree that enhancing the specificity of the conclusion, particularly in relation to symptom management and the strength of supporting evidence, adds meaningful value to the manuscript.

In response, we have revised the conclusion to include a summary statement that highlights common long COVID symptoms and their self-care approaches that are supported by clinical guidelines and expert consensus [please refer, section 7 (lines 415-422)]. This addition aims to provide clearer guidance for healthcare professionals and improve the practical utility of our review.

At the same time, as discussed in Section 4, the treatment of long COVID remains inherently complex due to its heterogeneous symptom profile and multifactorial pathophysiology. The lack of approved pharmacologic treatments and the limited availability of formal guidelines from major health authorities constrain the strength of evidence supporting specific interventions. While Section 4 and Table 1 offer a comprehensive overview of nonprescription options, the current evidence base does not permit formal grading of recommendations in many cases. We believe our approach presents a clear and balanced synthesis of available evidence without introducing contradictions or overstating certainty.

  1. Also in the conclusion section, no specific guidance is offered for healthcare professionals beyond the general need for further research.

Author response:

We sincerely thank the reviewer for this valuable observation, which has greatly contributed to enhancing the clarity and practical relevance of our manuscript. We acknowledge that the original conclusion lacked specific guidance for healthcare professionals, and we appreciate the opportunity to address this important gap.

In response, we have revised the conclusion to include targeted recommendations for clinical practice, grounded in the evidence presented throughout the manuscript (please refer to section 7, lines 424–430).

We believe these additions significantly improve the utility of the review for clinical audiences and strengthen its contribution to guiding symptom management in long COVID.

  1. I hope these comments are helpful as you revise your manuscript.

Author response:

We appreciated your suggestion, which contributed to enhancing both the structure and readability of the paper. We are grateful for your contribution to improving the overall quality and applicability of the manuscript.

THANK YOU!

Round 2

Reviewer 3 Report

Comments and Suggestions for Authors

Dear Authors,

Thank you for addressing my previous comments and for the revisions made to the manuscript. After reviewing the updated version, I believe that the clarifications and adjustments provided have significantly improved the clarity and rigor of the work. I have no further comments at this stage.

Yours sincerely,